# DCRM-ViT: Domain-Conditioned Residual Modulation for Multi-Domain Vision Transformers

## Abstract

Medical imaging presents significant challenges due to acoustic shadows, motion blur, and indistinct boundaries. Addressing these issues is crucial for improving diagnostic accuracy. Many conventional vision models require extensive fine-tuning on task-specific data and often lose generalizability to natural-image domains. We propose `DCRM-ViT`, a domain-conditioned residual modulation framework for Vision Transformers that preserves general-vision capability while adapting to diverse domains. `DCRM-ViT` keeps the backbone frozen and augments each block with a lightweight Residual Modulation Block (`RMB`) whose parameters are synthesized per sample by a Domain Router (`DR.`) and Parameter Synthesizer Network (`PSN`). The router outputs soft domain weights from input features, whereas the synthesizer maps these weights to low-rank residuals that modulate selected projections and, optionally, add a domain-aware bias to attention. Crucially, we learn routing and modulation via a bi-level optimization scheme: a short inner loop adapts `RMB` parameters to task supervision, while an outer loop updates `DR.`, `PSN`, and `RMB` initializations/step sizes so the synthesized residuals generalize across medical and natural domains. Across fine-grained classification (Food101, SUN397, Stanford Cars) and medical segmentation (ultrasound, CT, MRI), `DCRM-ViT` improves over strong baselines while using modest trainable compute. The ablation studies confirmed the benefits of our architectural enhancements, showing improved performance and adaptability. The results demonstrate `DCRM-ViT`'s potential to offer high diagnostic performance with reduced computational overhead of using 4.7 GFLOPs and 0.3 training min/epoch. Our code will be publicly available upon acceptance.

## 1 Introduction

Modern vision systems increasingly need a single model that can operate across different regimes like: (i) general natural imagery, which supports broad visual competence, quality control, and upstream data services; and (ii) medical, where decisions are high–stakes and inputs are affected by challenges like speckle, shadows, and device variability. As illustrated in Figure 1, fetal scans are characterized by low signal-to-noise ratios, significant speckle noise, acoustic shadows that obscure anatomical detail, and frequent variations in fetal pose and scale. Such domain-specific complexities require fine-grained visual understanding and robustness to modality-specific artifacts, capabilities that general vision models typically lack. The practical objective is to gain robustness in the medical domain without eroding performance on natural images (Guan & Liu, 2021).

Conventional methods partially address this objective, as full fine-tuning can improve accuracy in the domain, but often narrows competence and increases training cost (Davila et al., 2024). Unsupervised domain adaptation presumes access to target data and relies on alignment objectives that may be brittle across scanners, sites, or protocols (Zhou et al., 2025). The adaptation in test time alters either parameters or statistics during inference (Xiao & Snoek, 2024), thereby complicating the validation process. Taken together, these limitations motivate a training strategy that improves medical robustness while explicitly preserving general–image capability, and that keeps inference simple and predictable.

A key observation is that the variation between medical and natural images is continuous and not purely categorical (Konz & Mazurowski, 2024). This continuity suggests replacing heavy and static domain-wise adjustments with minimal, input-conditioned corrections that act only where beneficial and only by the amount required. Let $f_{\theta_0}$ be a pre-trained Vision Transformer (ViT) with frozen parameters $\theta_0$. Given a labeled training data $\{(x_i, y_i)\}_{i=1}^N$ obtained from a mixture $\mathcal{D} = \bigcup_{k=1}^K \mathcal{D}_k$ of domains. The goal is to learn an input-conditioned residual modulation mechanism that, for any test input $x$, produces small, low-rank corrections applied to a subset of encoder projections, with no gradient-based updates for inference.

In this paper, we target a solution that: *(i)* maintains a single deployable model for both medical and natural images, *(ii)* preserves fixed, fast inference with no test–time updates, *(iii)* respects tight memory/latency constraints, and *(iv)* remains stable across different modalities. For this purpose, we introduce a method named **D**omain-**C**onditioned **R**esidual **M**odulation for **Vi**sion **T**ransformers (**DCRM-ViT**): a frozen ViT having four minimal components that operate only where needed and only as much as needed: *(i)* a **Domain Router (DR.)** that takes input features and emits soft domain weights (Table 8 of Appendix), *(ii)* a tiny **P**arameter **S**ynthesizer **N**etwork (**PSN**) that maps those weights to per-sample, low-rank parameters, *(iii)* **R**esidual **M**odulation **B**locks (**RMB**) placed at selected projections (value and MLP streams) inside the frozen ViT, and *(iv)* a **D**omain-**A**ware **B**ias (**DAB**) that perturbs attention logits with negligible overhead.

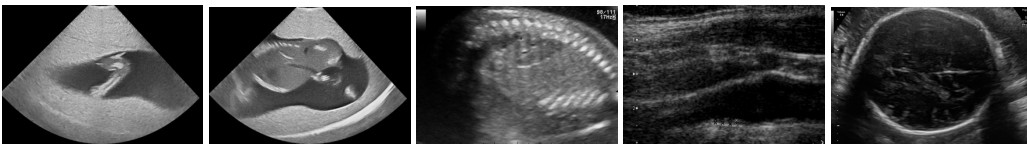

(a) Motion-induced blur    (b) Varying fetal pose and scale    (c) Blurred indistinct anatomy boundaries    (d) Speckle noise and low SNR    (e) Acoustic shadowing from the skull

Figure 1: Examples of key challenges in fetal ultrasound imaging.

We retain a frozen, pretrained encoder and attach small, input-conditioned residual modulators that make minimal corrections to a subset of projections. This preserves fast, update-free inference while providing the flexibility needed to handle ultrasound artifacts. The **DR.** module enables **DCRM-ViT** to perform domain-aware feature adaptation on a per-sample basis. It comprises four components: a **G**ating **C**hannel **U**nit (GCU), a **D**omain-**A**ware **L**ay**er** (DALer) unit, a domain classifier, and a **P**arameter **S**ynthesizer **N**etwork (PSN). Furthermore, **RMB** modules are inserted into each transformer block and specialize in learning domain-specific representations that reshape self-attention and feedforward activations to become sensitive to domain-specific artifacts such as noise, acoustic shadows, and low-contrast edges, without modifying the pre-trained backbone weights.

Now, improving one domain can silently harm the other. In addition, we frame our model training under a bi-level optimization strategy. In the inner loop, we fine-tune the parameters **DR.** with gradient steps to learn visual representations of the data for each task, while in the outer loop, we update the parameters of the **RMB** module (along with **DR.** initializations and inner loop learning rate) using domain characteristic examples, teaching the **PSN** unit

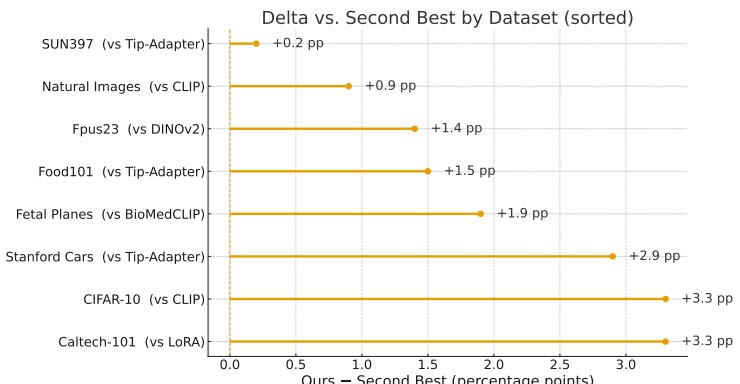

Figure 2: Percentage-point improvement of Ours over the second-best baseline, sorted by gain.

to produce domain-aware parameters that work across images from the medical and natural domains. This nested optimization cleanly separates domain-level meta-parameters from task-level representation learning. To our knowledge, this is the first unified framework for medical (ultrasound + CT +

MRI) analysis that explicitly disentangles domain adaptation and task specialization through separate, jointly optimized modules for domain-aware parameterization and task-specific representation learning.

**Contributions:** *(1)* We introduce the `DCRM-ViT` framework, the first end-to-end transformer architecture that unifies per-sample, domain-aware feature adaptation for both medical and natural image inputs within a single backbone. *(2)* We develop a domain-aware `DR.` module that dynamically adjusts the parameters of the `RMB` module based on the image domain, optimizing the model's performance for general or medical imagery. *(3)* Our extensive experimental study on the medical domain, including `ultrasound + CT + MRI`, alongside natural imagery, reveals that `DCRM-ViT` provides notable gains over CLIP, MAE, and DINOv2 in both zero-shot and cross-domain applicability, as can also be seen in Figure 2. In addition, we also present comprehensive results on segmentation benchmarks, showing consistent ultrasound, CT, and MRI gains with bounded impact on natural-image performance.

## 2 RELATED WORK

**Foundational Models in Medical Imaging.** Recent advancements in medical imaging include MedCLIP (Wang et al., 2022) and BiomedCLIP (Zhang et al., 2023a), which improve tasks such as zero-shot prediction and image-text retrieval. These models demonstrate significant diagnostic improvements through training on large datasets, highlighting the efficacy of vision-language processing in medical applications. Similarly, conversational AI models such as LLaVA-Med (Li et al., 2024) and XrayGPT (Thawkar et al., 2023) enhance interactions between medical professionals and AI systems, facilitating dialogues for querying, explaining, and instructing medical images. These techniques integrate AI into clinical workflows, improving decision-making and patient care by providing immediate, relevant information.

**Low-Rank Adaptation in Deep Learning.** Parameter-efficient fine-tuning (Fu et al., 2023) uses small, trainable modules that can be inserted into pre-trained network architectures. They offer an efficient way to adjust models across different domains without extensive retraining. These modules enable fine-tuning on specific tasks while preserving the original network's weights, thus maintaining broad generalization capabilities (Houlsby et al., 2019). Such architectures have been widely explored in computer vision and NLP to achieve parameter-efficient fine-tuning. In computer vision, adapters (Gao et al., 2024; Sung et al., 2022) have proven to be effective in handling visual domains with minimal parameter updates, demonstrating their utility when data is scarce or tasks are highly specific (Rebuffi et al., 2017). In NLP, (Houlsby et al., 2019) introduced adapter modules that enable task-specific adaptation of large language models with minimal parameter updates. Building on this, (Hu et al., 2022) proposed Low-Rank Adaptation (LoRA), which uses low-rank matrices to efficiently fine-tune large-scale Transformer models with far fewer trainable parameters. In computer vision, (Jia et al., 2022) proposed Visual Prompt Tuning (VPT), which adapts pre-trained vision models to downstream tasks using learnable prompts instead of full fine-tuning. AdapterFusion (AF) (Pfeiffer et al., 2020) assigns a distinct module to each task, enabling task-specific specialization. In contrast, we adopt a shared module architecture across natural and medical domains, reducing parameters and enhancing generalization. Unlike AF, which suits multi-task settings with distinct, known tasks, our approach targets a shared classification problem where unified representations are both practical and beneficial. Meta-adaptive control using `DR.` enables input-aware modulation without relying on task-specific modules.

**Meta-Learning for Dynamic Adaptation.** Meta-adaptation refers to the ability of a model to dynamically adjust its internal components or parameters based on the input context, such as domain, task, or data characteristics. It is inspired by meta-learning ("learning to learn"), but instead of training separate modules for each domain or task, a controller (often a lightweight neural network) generates or selects module parameters on the fly depending on the input. Model-Agnostic Meta-Learning (MAML) exemplifies this by preparing models to adjust quickly using a few examples (Finn et al., 2017), which is vital for scenarios such as medical diagnostics where data may be scarce. A comprehensive review (Hospedales et al., 2021) underscores the broad applications of meta-learning, from classification to complex autonomous decisions, pointing to future dynamic adaptation techniques. A recent study (Song et al., 2023) introduces a meta-adaptive approach that dynamically adjusts parameters to enhance the performance of vision-language models in a few-shot

learning scenarios. They perform gated multi-head attention over support images to fine-tune CLIP's (Radford et al., 2021) text embeddings for few-shot vision–language tasks. Despite these advances, the use of such techniques in medical imaging, particularly ultrasound analysis, remains largely underexplored. Our work builds on these foundations by proposing a unified framework named **DCRM-ViT**, combining domain-aware and task-specific modules tailored to the unique challenges of medical imaging, such as acoustic shadows, motion blur, and indistinct boundaries.

## 3 METHODOLOGY

In this section, we describe the design of **DCRM-ViT** by detailing its core modules and our two-stage training objective. We begin with an architectural overview, then dive into the structure of each component, and finally present the optimization strategy. **DCRM-ViT** augments a standard Vision Transformer backbone by initially embedding **DR.** and then an **RMB** module into every transformer block as shown in Figure 3, which enables per-sample, domain-aware feature adaptation for both medical and natural-image inputs.

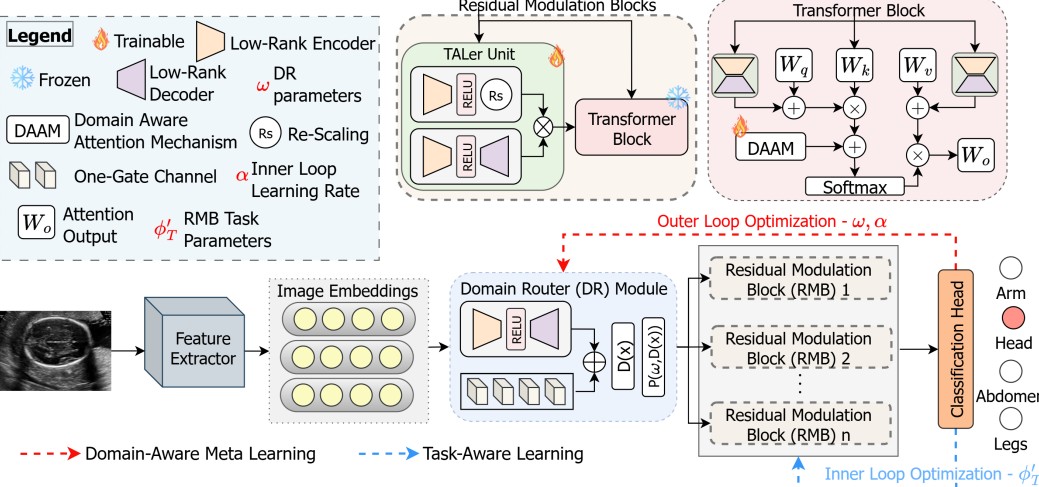

Figure 3: **DCRM-ViT** processes images through a backbone to generate image embeddings, which are then adapted by a **DR.** module. This module uses outputs from a domain classifier to dynamically create **DR.** parameters via the **PSN** unit. These parameters direct transformations within **RMB** modules, preparing the data for a transformer block with domain-aware attention. The features are refined further through TALer and transformer blocks before reaching the classification head that determines the image class.

**Residual Modulation Blocks:** The **RMB** module in the **DCRM-ViT** model modifies the transformer architecture to enhance feature processing for medical imaging tasks. Adjusting the dimensions of input features enables precise data manipulation, focusing on extracting subtle diagnostic details crucial for medical imaging. This technical enhancement optimizes the model's ability to accurately identify fine-grained anatomical structures and tissue boundaries, as shown in Figure 5 of the appendix. Each **RMB** module contains two main components: the **T**ask-**A**ligned **L**ayer (TALer) unit and the transformer block. Each **RMB** module's input is fed to the TALer layer and the transformer block. The output of the transformer block serves as the input for the subsequent block. The detailed descriptions of each component are given below:

**TALer unit.** In each **RMB** module, the TALer unit is integrated with the transformer block of the model to enable efficient domain-specific fine-tuning. These layers are designed to modify the feature representations with minimal computational overhead, as shown in Table 5, preserving the original model's broad generalization capabilities. Each TALer unit consists of a low-rank encoder, a non-linear activation function (ReLU), and a low-rank decoder. We also integrated the rescale variable that helps capture the visual features at different scales, as the medical object can vary in

size (lesion vs. fetal anatomical structures). Mathematically, for an input feature vector $\mathbf{h} \in \mathbb{R}^d$, the transformation $\mathbf{h}'$ applied by a TALer unit is shown in Equation 1:

$$\mathbf{h}' = \mathbf{r} \cdot \sigma(\mathbf{W}_{DS} \cdot \mathbf{h} + \mathbf{b}_{DS}) \odot (\mathbf{W}_{US} \cdot \sigma(\mathbf{W}_{DS} \cdot \mathbf{h} + \mathbf{b}_{DS}) + \mathbf{b}_{US}) \tag{1}$$

where r is the rescaling factor, $\mathbf{W}_{DS} \in \mathbb{R}^{d' \times d}$ is the weight matrix of the low-rank encoder, $\mathbf{W}_{US} \in \mathbb{R}^{d \times d'}$ is the weight matrix of the low-rank decoder, $\mathbf{b}_{DS} \in \mathbb{R}^{d'}$ and $\mathbf{b}_{US} \in \mathbb{R}^d$ are the bias terms, and $\sigma$ represents the ReLU activation function. The dimensionality $d$ is the original feature dimension, while $d'$ is the reduced dimension within the TALer layer. As the input image progresses through the TALer unit, the low-rank encoder first reduces the dimensionality of the input features, making subsequent computations more efficient. Finally, the low-rank decoder restores the features' original dimensionality, ensuring that the modified representations are compatible with subsequent layers in the attention mechanism model.

**Transformer Block.** The attention mechanism in **DCRM-ViT** is based on the self-attention mechanism used in Transformers (Dosovitskiy et al., 2020). In the **DCRM-ViT** model, the transformer blocks are designed to process complex image features efficiently, utilizing a modified self-attention mechanism that integrates specialized TALer units. These TALer units adjust the projection layers for queries (Q) and values (V) to enhance the model performance and computational efficiency. The transformation vectors can be described mathematically as $Q = \tilde{W}_q \gamma = W_q \gamma + \mathbf{h}'_{\mathbf{q}} \gamma$ and $V = \tilde{W}_v \gamma = W_v \gamma + \mathbf{h}'_{\mathbf{v}} \gamma$.

Here, $W_q$ and $W_v$ are the original, frozen projection layers for queries and values, respectively. $\mathbf{h}'_{\mathbf{q}}$ and $\mathbf{h}'_{\mathbf{v}}$ are trainable parameters introduced by TALer units, allowing the model to learn domain-specific feature representations, while $\gamma$ is the input vector. Here, the key projection remains unchanged to preserve the integrity of the attention mechanism, which is written as $K = W_k \gamma$. This consistency ensures that attention scores reflect genuine relationships within the data without alteration by the TALer units. Furthermore, we also introduce a domain-aware attention mechanism using domain-aware biases (**DAB**) within the transformer blocks to capture domain-specific contextual relationships. The self-attention computation is modified to include domain-specific attention biases as presented in Equation 2:

$$\text{Attention}(\mathbf{Q}, \mathbf{K}, \mathbf{V}) = \text{softmax}\left(\frac{\mathbf{Q}\mathbf{K}^T}{\sqrt{d_k}} + \mathbf{B}_d\right) \mathbf{V} \tag{2}$$

where $\mathbf{B}_d$ corresponds to $\mathbf{B}_d = p_{\text{medical}} \mathbf{B}_{\text{medical}} + p_{\text{natural}} \mathbf{B}_{\text{natural}}$, and is a domain-specific bias matrix, $d_k$ is the dimensionality of the keys, $\mathbf{B}_{\text{medical}}$ and $\mathbf{B}_{\text{natural}}$ are learned bias matrices for the medical and natural domains, respectively. This modification allows attention mechanism to focus differently based on domain probabilities, improving the model's ability to capture nuanced domain-specific patterns.

**Domain Router Module:** The **DR.** module dynamically adjusts the TALer units' parameters based on the input image's domain. This design is inspired by prior work (Malik et al., 2023; Bansal et al., 2022), which employed meta-learning techniques to modulate standard architectures for domain adaptation. **DR.** comprises **D**omain-**A**ware **Lay**er (DALer) unit, which contains two types of layers named domain-aware contraction and expansion layer, respectively, along with a non-linear activation. However, we also incorporate a parallel gate channel, consisting of $1 \times 1$ convolutional layers to obtain valuable features and concatenate later, as shown in the **DR.** module box of Figure 3. The gate channel applies 1x1 convolutional layers to the initial image embeddings. These layers are adept at transforming the feature space without altering the depth of the embeddings, allowing for precise manipulation of the spatial features.

This transformation is crucial for extracting refined, domain-specific features from the embeddings, which are essential for subsequent processing steps. The output of this convolution is mathematically represented as $\mathbf{g} = \text{Conv}_{1 \times 1}(\mathbf{x}; \theta_g)$, where $\theta_g$ are the trainable parameters of the convolutional layer, which forms a set of gate-processed features. Following feature extraction, the gate-processed features ($\mathbf{g}$) are concatenated with the original embeddings. This enriched set of features combines the original data with newly emphasized domain-specific attributes, creating a composite input that feeds into the **RMB** modules. This **RMB** module is driven by the need for a versatile model capable of handling various imaging modalities and conditions. It adjusts the parameters of the TALer units, ensuring optimal model performance whether processing 'medical' or 'natural' images. In general, it

comprises a domain classifier $D$, which determines the domain of the input image $\mathbf{x}$. The domain classifier $D$ uses a softmax layer to output a probability distribution over the possible domains (e.g., medical, natural) and can be written as $D(\mathbf{x}) = [p_{\text{medical}}, p_{\text{natural}}]$, where $p_{\text{medical}}$ and $p_{\text{natural}}$ represent the probabilities that the input image $\mathbf{x}$ belongs to the medical or natural domain, respectively.

**Enhanced `DR.` module with Dynamic Parameter Generation.** To further elevate the adaptability of the `DCRM-ViT` model across diverse domains, we introduce a novel *dynamic adapter parameter generation* mechanism within the `DR.` module. Instead of relying on fixed parameters determined by hard domain classification, we employ a Parameter Synthesizer Network (`PSN`) unit inspired by (Ha et al., 2016) that generates `RMB` parameters conditioned on the input image features and domain probabilities. Here, the `PSN` unit is a fully connected neural network that outputs the weight and bias values for the TALer units. The size of this network's output layer corresponds to the total number of parameters in these units. This approach allows continuous adaptation to the input domain characteristics.

Specifically, the `PSN` unit P takes the initial image embeddings $\mathbf{x}$ and the domain probabilities $D(\mathbf{x})$ as input, outputting the parameters for the TALer units as $\theta_A = P(\mathbf{x}, D(\mathbf{x}); \theta_P)$ where $\theta_P$ denotes the parameters of the `PSN` unit, and $D(\mathbf{x}) = [p_{\text{medical}}, p_{\text{natural}}]$ are the domain probabilities obtained from the domain classifier. This conditioning allows the `PSN` unit to generate parameters influenced by the degree to which the input image belongs to each domain, providing smooth interpolation between domain-specific parameters.

**Image Encoder:** The input image is first divided into non-overlapping patches of size $p \times p$. These patches are then linearly embedded into vectors and augmented with positional embeddings to retain spatial information. Given an image $\mathbf{x} \in \mathbb{R}^{H \times W \times C}$ with height $H$, width $W$, and channels $C$, the image is divided into $N_p = \frac{HW}{p^2}$ patches. Each patch $\mathbf{p}_i \in \mathbb{R}^{p \times p \times C}$ is projected to a vector $\mathbf{e}_i \in \mathbb{R}^D$ using a linear layer, where $D$ is the embedding dimension. The set of patch embeddings is given by $\mathbf{E} = \{\mathbf{e}_0, \mathbf{e}_1, \ldots, \mathbf{e}_{N_p-1}\}$, where $\mathbf{e}_0$ is a learnable class token added to aggregate global image information. The embeddings and positional embeddings are fed into `DR.` and a series of `RMB` modules, generating new token representations that encode the image features.

**Training Strategy.** The enhanced `DR.` module effectively optimizes the `PSN` unit and primary model. We employ a joint training approach where the domain classifier, `PSN` unit, and the main modules are trained simultaneously. The overall loss function is a combination of the target classification loss, domain classification loss, and the regularization term and is shown as $\mathcal{L}_{\text{total}} = \mathcal{L}_{\text{cls}} + \beta \mathcal{L}_{\text{domain}} + \mathcal{L}_{\text{reg}}$, where $\mathcal{L}_{\text{cls}}$ is the standard cross-entropy loss for the classification task, $\mathcal{L}_{\text{domain}}$ is the loss for the domain classifier (e.g., cross-entropy loss between predicted and true domain labels), and $\beta$ is a weighting factor that balances the importance of domain classification in the overall training process.

**Learning Objective.** Bi-level optimization is a strategy where two levels of optimization tasks are solved concurrently. Inspired by the work in (Finn et al., 2017), our model adaptively uses an MAML-based framework to fine-tune parameters across different tasks. The upper level (outer loop) optimizes the meta-learning parameters for domain distinction (medical or natural domain), while the lower level (inner loop) focuses on task-specific (e.g., fetal head, abdomen classification or lesion segmentation) model parameters. This optimization is critical for `DCRM-ViT` because it cleanly disentangles the learning of domain-level meta-parameters (the `DR.` weights $\omega$ and initial `RMB` states $\phi$) from the task-specific fine-tuning of `RMB`. Without this nested scheme, the `PSN` unit would receive conflicting gradient signals—simultaneously trying to serve all tasks and domains—leading to suboptimal domain inference and slower adaptation. Thus, the bi-level formulation provides a principled way to meta-learn the best initialization and parameter-generation strategy, enabling `DCRM-ViT` to dynamically synthesize per-sample calibration kernels that recover ultrasound artifacts and preserve general-vision features. Building on this foundation, we introduce an enhanced training strategy integrating dynamic `RMB` parameter generation, soft parameter sharing, and a domain-aware attention mechanism. Here, the objective for each individual task $T$ is shown in Equation 3:

$$\phi_T \leftarrow \arg\min_\phi \mathcal{L}_T(\theta, \phi, \omega; \mathcal{D}_T) \tag{3}$$

where $\mathcal{L}_T$ represents the loss for task $T$, and $\mathcal{D}_T$ denotes the dataset for task $T$. The parameters $\omega$ are not fine-tuned for individual task $T$ but instead optimized for the target task, i.e., domain classification here. Thus, $\omega$ is optimized to improve `RMB` fine-tuning, as shown in objective function Equation 4:

$$\min_\omega \mathbb{E}_T \left[ \mathcal{L}_T(\theta, \phi_T, \omega; \mathcal{D}_T) \right] \tag{4}$$

A few steps of gradient descent are used to solve this nested minimization. This optimization problem was inspired by the approach used in (Bansal et al., 2022), which also involves an episodic framework where each episode optimizes different objectives. The initial objective uses domain characteristic data $\mathcal{D}_T^{\text{dom}}$, whereas the other objective uses task-specific data $\mathcal{D}_T^{\text{task}}$. Here, $\mathcal{D}_T^{\text{dom}}$ is used for outer loop optimization, and $\mathcal{D}_T^{\text{task}}$ is used for inner loop optimization.

**Inner Loop Optimization (Task-Specific Adaptation).** In the inner loop, for each task $T$, we fine-tune the **RMB** parameters $\phi$ using multiple steps of gradient-descent based on the $\mathcal{D}_T^{\text{task}}$ dataset. The update rule is: $\phi'_T \leftarrow \phi - \alpha \nabla_\phi \mathcal{L}_T(\theta, \phi, \omega; \mathcal{D}_T^{\text{task}})$. Here, $\alpha$ is the learning rate for the inner loop, $\theta$ denotes the pre-trained transformer parameters (kept frozen), $\omega$ represents the parameters of the **DR.** module, and $\mathcal{L}_T$ is the task-specific loss function. With the introduction of the **PSN** unit $P$ for dynamic parameter generation, the **RMB** parameters $\phi$ are now functions of $\omega$ and the domain probabilities $D(\mathbf{x})$: $\phi = P(\omega, D(\mathbf{x}))$. This means that the **RMB** parameters are dynamically generated based on the **DR.** parameters and the estimated domain probabilities, allowing the model to adapt more effectively to task-specific features.

**Outer Loop Optimization (Meta-Learning).** In the outer loop, we optimize the **DR.** module parameters ($\omega \leftarrow \omega - \beta \nabla_\omega \mathbb{E}_T \left[ \mathcal{L}_T(\theta, \phi'_T, \omega; \mathcal{D}_T^{\text{dom}}) \right]$), the initial **RMB** parameters ($\phi \leftarrow \phi - \beta \nabla_\phi \mathbb{E}_T \left[ \mathcal{L}_T(\theta, \phi'_T, \omega; \mathcal{D}_T^{\text{dom}}) \right]$), and the learning rate ($\alpha \leftarrow \alpha - \beta \nabla_\alpha \mathbb{E}_T \left[ \mathcal{L}_T(\theta, \phi'_T, \omega; \mathcal{D}_T^{\text{dom}}) \right]$) using the $\mathcal{D}_T^{\text{dom}}$ dataset. The initial **RMB** parameters help avoid random initialization. Here, $\beta$ is the learning rate for the outer loop, ensuring that the **DR.** parameters, **RMB** initialization, and fine-tuning learning rates are all optimized for effective generalization across diverse tasks. This dual loop approach ensures that both $\phi$ and $\omega$ are iteratively refined, optimizing the model's adaptability and effectiveness in applying learned knowledge to new and varied tasks. Using this bi-level optimization approach, the **DCRM-ViT** model can effectively learn to adapt to new tasks with minimal data, leveraging the dynamic adjustment capabilities of the **DR.** module to optimize performance across different domains. The pseudo algorithm is also given in A.9.

Table 1: Combined **fine-tuning** performance across fetal (Fpus23, Fetal Planes), natural imagery (CIFAR-10, Caltech101, Natural Images), and fine-grained datasets (Food101, SUN397, S-Cars).

| Model | Medical | | | | Standard Datasets | | | | | | Fine-Grained Datasets | | | | | |
| | Fpus23 | | Fetal Planes | | CIFAR-10 | | Caltech101 | | Natural Images | | Food101 | | SUN397 | | S-Cars | |
| | Acc. (%) | F1 | Acc. (%) | F1 | Acc. (%) | F1 | Acc. (%) | F1 | Acc. (%) | F1 | Acc. (%) | F1 | Acc. (%) | F1 | Acc. (%) | F1 |
|---|---|---|---|---|---|---|---|---|---|---|---|---|---|---|---|---|
| MAE | 57.7±1.5 | 0.58±0.04 | 81.5±0.7 | 0.82±0.03 | 86.5±0.6 | 0.87±0.02 | 81.4±1.2 | 0.82±0.03 | 78.2±1.1 | 0.79±0.04 | 94.7±0.2 | 0.95±0.01 | 80.3±0.4 | 0.80±0.01 | 90.4±0.3 | 0.90±0.01 |
| DINOv2 | 59.3±1.2 | 0.60±0.03 | 87.8±0.5 | 0.85±0.02 | 87.9±0.5 | 0.88±0.01 | 83.1±1.0 | 0.83±0.01 | 80.5±1.0 | 0.81±0.03 | 95.1±0.3 | 0.95±0.01 | 80.6±0.4 | 0.81±0.01 | 90.8±0.3 | 0.91±0.01 |
| CLIP | 61.6±1.0 | 0.62±0.02 | 42.2±2.0 | 0.45±0.06 | 88.4±1.4 | 0.88±0.01 | 84.1±0.9 | 0.84±0.02 | 81.9±0.9 | 0.82±0.02 | 93.1±0.4 | 0.93±0.01 | 78.4±0.6 | 0.78±0.02 | 88.7±0.5 | 0.89±0.01 |
| Tip-Adapter | 60.3±1.3 | 0.61±0.03 | 85.5±0.6 | 0.86±0.01 | 87.0±0.7 | 0.87±0.04 | 82.5±1.1 | 0.83±0.02 | 79.5±1.2 | 0.80±0.03 | 94.2±0.3 | 0.94±0.01 | 79.3±0.5 | 0.79±0.01 | 89.9±0.4 | 0.90±0.01 |
| AdaptFormer | 62.1±0.9 | 0.63±0.01 | 88.0±0.5 | 0.88±0.02 | 87.8±0.6 | 0.88±0.01 | 83.9±0.8 | 0.84±0.02 | 81.0±1.0 | 0.82±0.02 | 93.6±0.4 | 0.94±0.01 | 78.8±0.6 | 0.79±0.02 | 89.4±0.5 | 0.89±0.01 |
| LoRA | 63.0±0.8 | 0.65±0.02 | 88.3±0.4 | 0.90±0.01 | 88.1±0.5 | 0.88±0.01 | 84.0±0.7 | 0.84±0.02 | 82.0±0.8 | 0.82±0.02 | 94.5±0.3 | 0.95±0.01 | 79.8±0.4 | 0.80±0.01 | 90.1±0.3 | 0.90±0.01 |
| **DCRM-ViT** | **63.4±0.7** | **0.69±0.02** | **89.3±0.3** | **0.90±0.01** | **89.2±0.4** | **0.89±0.01** | **85.8±0.6** | **0.85±0.02** | **82.5±0.5** | **0.83±0.02** | **95.7±0.2** | **0.96±0.01** | **81.3±0.3** | **0.81±0.01** | **91.5±0.2** | **0.92±0.01** |

**Zero-Shot Transformation:** To enable zero-shot classification capabilities in models such as MAE (He et al., 2022), DINOv2 (Oquab et al., 2023), **DCRM-ViT**, and others, which are typically not designed for direct zero-shot learning like CLIP (Radford et al., 2021), a refined transformation approach is employed based on the work of (Moayeri et al., 2023). This transformation is done to evaluate the performance of these models against CLIP, MedCLIP Wang et al. (2022), UniMed-CIP (Khattak et al., 2024), or similar models in a zero-shot manner as well. This strategy involves aligning the vision-based feature representations of these models with the integrated text-image embedding space used by CLIP through a learned transformation matrix $W$ and a bias vector $b$, facilitating semantic alignment that enables zero-shot learning. More details are in A.2

Table 2: Zero-shot performance on medical (fetal) datasets

| Dataset | Metric | MAE | DINOv2 | CLIP | MedCLIP | UniMedCLIP | BioMedCLIP | Tip-Adapter | AdaptFormer | LoRA | **DCRM-ViT** |
|---|---|---|---|---|---|---|---|---|---|---|---|
| Fpus23 | Acc. (%) | 17.1±1.85 | 28.9±1.34 | 17.9±1.90 | 22.1±2.10 | 26.5±1.20 | 23.4±1.95 | 18.5±1.80 | 19.3±1.70 | 20.2±1.50 | **30.3±2.50** |
| | F1 | 0.19±0.04 | 0.29±0.05 | 0.18±0.03 | 0.20±0.045 | 0.27±0.05 | 0.26±0.04 | 0.19±0.035 | 0.20±0.03 | 0.21±0.025 | **0.31±0.06** |
| Fetal Planes | Acc. (%) | 28.2±1.20 | 29.4±1.50 | 27.4±1.25 | 29.9±1.40 | 30.4±0.80 | 30.9±1.35 | 27.8±1.15 | 28.4±1.10 | 29.3±1.05 | **32.8±2.00** |
| | F1 | 0.31±0.03 | 0.27±0.04 | 0.30±0.025 | 0.25±0.035 | 0.30±0.03 | 0.29±0.03 | 0.28±0.025 | 0.29±0.02 | 0.30±0.015 | **0.33±0.05** |

## 4 RESULTS

**Fine-tuning Results:** We explored the fine-tuning capabilities of MAE (He et al., 2022), DINOv2 (Oquab et al., 2023), MedCLIP Wang et al. (2022), BioMedClip (Zhang et al., 2023b), UniMed-

CIP (Khattak et al., 2024), and CLIP (Radford et al., 2021), as well as the low-rank methods like Tip-Adapter (Zhang et al., 2021), AdaptFormer (Chen et al., 2022), LoRA (Hu et al., 2021), and our **DCRM-ViT** model across both specialized medical and varied natural imagery datasets. As shown in Table 1, **DCRM-ViT** outperformed its counterparts, notably excelling on the fetal datasets Fpus23 (Prabakaran et al., 2023) and Fetal Planes Burgos-Artizzu et al. (2020), achieving accuracies of 63.4% and 89.3%, respectively. When extended to natural imagery datasets, **DCRM-ViT** again demonstrated superior performance, achieving the highest accuracy and F1 scores among the models tested, as seen in Table 1.

Table 3: Zero-shot performance on natural-imagery datasets

| Dataset | Metric | MAE | DINOv2 | CLIP | MedCLIP | UniMedCLIP | BioMedCLIP | Tip-Adapter | AdaptFormer | LoRA | DCRM-ViT |
|---|---|---|---|---|---|---|---|---|---|---|---|
| **Standard Datasets** | | | | | | | | | | | |
| CIFAR-10 | Acc. (%) | $65.0 \pm 0.40$ | $68.5 \pm 0.45$ | $71.9 \pm 0.30$ | $66.7 \pm 0.50$ | $65.5 \pm 0.45$ | $67.0 \pm 0.48$ | $70.2 \pm 0.35$ | $69.8 \pm 0.40$ | $71.5 \pm 0.30$ | $\mathbf{75.2 \pm 0.50}$ |
| | F1 | $0.65 \pm 0.015$ | $0.68 \pm 0.02$ | $0.73 \pm 0.01$ | $0.67 \pm 0.015$ | $0.71 \pm 0.02$ | $0.68 \pm 0.02$ | $0.70 \pm 0.015$ | $0.70 \pm 0.015$ | $0.72 \pm 0.01$ | $\mathbf{0.74 \pm 0.02}$ |
| Caltech-101 | Acc. (%) | $62.0 \pm 0.35$ | $65.8 \pm 0.40$ | $68.92 \pm 0.50$ | $65.0 \pm 0.45$ | $64.3 \pm 0.42$ | $65.5 \pm 0.42$ | $67.9 \pm 0.30$ | $68.0 \pm 0.35$ | $69.2 \pm 0.25$ | $\mathbf{72.5 \pm 0.45}$ |
| | F1 | $0.62 \pm 0.02$ | $0.66 \pm 0.025$ | $0.72 \pm 0.02$ | $0.65 \pm 0.02$ | $0.71 \pm 0.02$ | $0.66 \pm 0.025$ | $0.68 \pm 0.02$ | $0.68 \pm 0.02$ | $0.69 \pm 0.015$ | $\mathbf{0.69 \pm 0.025}$ |
| Natural Images | Acc. (%) | $60.5 \pm 0.50$ | $64.0 \pm 0.55$ | $71.0 \pm 0.60$ | $63.8 \pm 0.55$ | $61.8 \pm 0.55$ | $64.0 \pm 0.50$ | $65.1 \pm 0.45$ | $66.3 \pm 0.40$ | $67.5 \pm 0.35$ | $\mathbf{71.9 \pm 0.60}$ |
| | F1 | $0.60 \pm 0.025$ | $0.64 \pm 0.03$ | $0.71 \pm 0.02$ | $0.63 \pm 0.025$ | $0.69 \pm 0.03$ | $0.64 \pm 0.03$ | $0.65 \pm 0.025$ | $0.66 \pm 0.02$ | $0.67 \pm 0.02$ | $\mathbf{0.71 \pm 0.03}$ |
| **Fine-Grained Datasets** | | | | | | | | | | | |
| Food101 | Acc. (%) | $82.3 \pm 0.8$ | $85.1 \pm 0.6$ | $88.2 \pm 0.7$ | $83.0 \pm 0.6$ | $82.6 \pm 0.6$ | $83.8 \pm 0.6$ | $89.4 \pm 0.5$ | $88.1 \pm 0.6$ | $88.6 \pm 0.5$ | $\mathbf{90.9 \pm 0.4}$ |
| | F1 | $0.82 \pm 0.020$ | $0.85 \pm 0.020$ | $0.88 \pm 0.020$ | $0.83 \pm 0.020$ | $0.83 \pm 0.020$ | $0.84 \pm 0.020$ | $0.89 \pm 0.015$ | $0.88 \pm 0.020$ | $0.89 \pm 0.015$ | $\mathbf{0.91 \pm 0.015}$ |
| SUN397 | Acc. (%) | $55.7 \pm 0.9$ | $57.8 \pm 0.7$ | $62.3 \pm 0.8$ | $56.2 \pm 0.7$ | $55.9 \pm 0.7$ | $56.8 \pm 0.7$ | $63.1 \pm 0.7$ | $62.2 \pm 0.6$ | $62.0 \pm 0.6$ | $\mathbf{63.3 \pm 0.5}$ |
| | F1 | $0.56 \pm 0.030$ | $0.58 \pm 0.025$ | $0.62 \pm 0.030$ | $0.56 \pm 0.025$ | $0.56 \pm 0.025$ | $0.57 \pm 0.025$ | $0.63 \pm 0.025$ | $0.62 \pm 0.020$ | $0.62 \pm 0.020$ | $\mathbf{0.63 \pm 0.020}$ |
| Stanford Cars | Acc. (%) | $64.2 \pm 0.7$ | $67.8 \pm 0.5$ | $69.4 \pm 0.6$ | $65.0 \pm 0.6$ | $64.5 \pm 0.6$ | $65.7 \pm 0.6$ | $71.1 \pm 0.5$ | $70.3 \pm 0.6$ | $70.0 \pm 0.5$ | $\mathbf{74.0 \pm 0.4}$ |
| | F1 | $0.64 \pm 0.020$ | $0.68 \pm 0.020$ | $0.69 \pm 0.020$ | $0.65 \pm 0.020$ | $0.65 \pm 0.020$ | $0.66 \pm 0.020$ | $0.71 \pm 0.015$ | $0.70 \pm 0.020$ | $0.70 \pm 0.015$ | $\mathbf{0.74 \pm 0.015}$ |

**Zero-shot Classification Results:** Our evaluation of zero-shot classification focused on models' performance across fetal and natural imagery datasets without specific training on those datasets. The models used knowledge transferred from related tasks. **DCRM-ViT** achieved the highest accuracies in the fetal category, as can be seen in Table 2, which shows its robustness in medical imaging. In natural imagery, **DCRM-ViT** performed exceptionally, achieving best performance as can be seen in Table 3. Other specialized models like MedCLIP (Wang et al., 2022) and BioMedCLIP (Zhang et al., 2023b), optimized for medical imaging, showed improvements in fetal datasets but did not surpass **DCRM-ViT** in both fetal and natural images. However, these models performed below the baseline CLIP in natural imagery datasets, indicating their focus on medical applications with adequate generalization capabilities.

Table 4: Cross-domain transfer after fine-tuning.

| Model | Tuned on Fetal → Tested on Natural | | | Tuned on Natural → Tested on Fetal | |
|---|---|---|---|---|---|
| | CIFAR-10 | Caltech101 | Natural Img. | Fpus23 | Fetal Planes |
| MAE | $56.4 \pm 1.2/0.55 \pm 0.05$ | $58.0 \pm 1.3/0.57 \pm 0.02$ | $54.2 \pm 1.4/0.53 \pm 0.05$ | $51.0 \pm 1.5/0.55 \pm 0.04$ | $62.7 \pm 1.1/0.61 \pm 0.02$ |
| DINOv2 | $58.1 \pm 1.0/0.57 \pm 0.03$ | $60.5 \pm 1.1/0.59 \pm 0.04$ | $56.7 \pm 1.2/0.55 \pm 0.04$ | $53.6 \pm 1.3/0.51 \pm 0.05$ | $64.9 \pm 0.9/0.63 \pm 0.03$ |
| CLIP | $60.2 \pm 0.9/0.59 \pm 0.02$ | $62.3 \pm 1.0/0.61 \pm 0.01$ | $58.9 \pm 1.1/0.57 \pm 0.03$ | $55.9 \pm 1.2/0.55 \pm 0.04$ | $67.1 \pm 0.8/0.66 \pm 0.02$ |
| **DCRM-ViT** | $63.7 \pm 0.8/0.62 \pm 0.01$ | $65.8 \pm 0.9/0.64 \pm 0.02$ | $61.5 \pm 0.9/0.60 \pm 0.01$ | $58.4 \pm 1.0/0.57 \pm 0.03$ | $70.2 \pm 0.7/0.69 \pm 0.01$ |

**Computational Overhead** Table 5 compares total and trainable parameters alongside training throughput and approximate time per epoch under a fixed, fair protocol (ViT-B/16 at $224 \times 224$, FP16, batch=128, single A100-40GB means over warmed runs).

Table 5: Parameter count, train throughput (images/s), and approximate training time per epoch.

| Model | Total Params (M) | Trainable Params (M) | Train Throughput (img/s) | Training time/epoch (min) |
|---|---|---|---|---|
| MAE | 86.0 | 86.0 | 220 | 0.9 |
| DINOv2 | 87.0 | 87.0 | 210 | 1.0 |
| CLIP | 123.0 | 123.0 | 205 | 3.0 |
| AdaptFormer (PET) | 89.0 | 4.6 | 298 | 0.5 |
| LoRA (PET) | 88.4 | 5.5 | 308 | 0.45 |
| **DCRM-ViT** (ours) | **90.3** | **3.3** | **335** | **0.3** |

**Cross-Domain Results:** Our cross-domain evaluation assessed the model's adaptability when applied outside its initial training domains. This analysis provided crucial insights into the transfer learning capabilities of each model. In Table 4, models initially fine-tuned on fetal datasets, then tested on diverse natural imagery datasets like CIFAR-10 and Caltech101, showed that **DCRM-ViT** was notably superior. Similarly, when models trained on natural imagery were tested on specialized fetal

datasets such as Fpus23 and Fetal Planes, `DCRM-ViT` maintained the highest performance. These results highlight `DCRM-ViT`'s robust generalization capabilities, facilitated by its advanced feature adaptation strategies that effectively leverage domain-general knowledge.

Table 6: Impact of architectural components (addition/removal) on `DCRM-ViT`.

| Dataset | Metric | DR. | Gates | Drop-Path | GELU | DAAM | Rescale | PEN | RMB | Meta Learning |
|---|---|---|---|---|---|---|---|---|---|---|
| FPUS23 | Acc. (%) | 58.2 ± 1.3 | 60.7 ± 1.6 | 62.5 ± 0.9 | 62.9 ± 0.8 | 60.1 ± 1.0 | 61.4 ± 1.2 | 59.1 ± 1.4 | 51.4 ± 1.1 | 58.8 ± 0.8 |
| | F1 | 0.59 ± 0.04 | 0.60 ± 0.05 | 0.62 ± 0.02 | 0.64 ± 0.03 | 0.60 ± 0.02 | 0.62 ± 0.04 | 0.58 ± 0.05 | 0.52 ± 0.03 | 0.57 ± 0.05 |
| Fetal Planes | Acc. (%) | 78.2 ± 0.8 | 86.0 ± 1.1 | 88.7 ± 0.7 | 88.9 ± 0.5 | 86.1 ± 0.6 | 84.5 ± 0.9 | 85.7 ± 1.3 | 74.5 ± 1.5 | 83.1 ± 1.2 |
| | F1 | 0.77 ± 0.03 | 0.86 ± 0.04 | 0.88 ± 0.02 | 0.89 ± 0.01 | 0.85 ± 0.02 | 0.85 ± 0.03 | 0.85 ± 0.05 | 0.72 ± 0.03 | 0.86 ± 0.04 |

**Segmentation Results** Although `DCRM-ViT` is introduced as a domain-conditioned, per-sample classifier, its domain-conditioned mechanism transfers directly to segmentation, which is the clinically actionable endpoint in many prenatal and cardiac workflows. Concretely, we keep the ViT encoder *frozen* and attach a shallow per-pixel decoder (a $1 \times 1$ classifier with lightweight upsampling), training only this head together with the `RMB` and `DR.` using Dice+cross-entropy losses. This preserves the backbone's general-vision features while allowing per-sample corrections that are critical in ultrasound and equally beneficial in CT/MRI (adapting to modality-specific intensity and texture statistics). Table 7b summarizes segmentation performance for *ultrasound* (BUS-UCLM, BUID, BUS-BRA) and *other medical modalities* (cardiac CT/MRI via ACDC and MMWHS). Across all six datasets, `DCRM-ViT` retaining a frozen backbone and applying domain-conditioned residual modulation consistently outperforms strong baselines like FetalCLIP (Maani et al., 2025), SAMUS (Lin et al., 2024), SAMed (Zhang & Liu, 2023), and U-Net (Ronneberger et al., 2015), while keeping the trainable/latency overhead modest. On BUS-UCLM, BUID, and BUS-BRA, `DCRM-ViT` improves by **+3.07** points on average over SAMUS. Beyond ultrasound, improving with **+2.23** point gain over SAMUS.

Table 7: (a) Ablation on epochs for FPUS23/FP (Acc.%); (b) Segmentation Dice on ultrasound and cardiac CT/MRI.

(a) Ablation results for epochs on FPUS23 and FP.

| Model | FPUS23 Dataset (Acc.%) | | | | FP Dataset (Acc.%) | | | |
|---|---|---|---|---|---|---|---|---|
| | 15 Epochs | 30 Epochs | 50 Epochs | 100 Epochs | 15 Epochs | 30 Epochs | 50 Epochs | 100 Epochs |
| CLIP | 59.3 ± 1.2 | 61.5 ± 0.9 | 60.2 ± 1.1 | 58.8 ± 1.4 | 39.7 ± 1.5 | 42.1 ± 1.3 | 41.0 ± 1.2 | 39.9 ± 1.6 |
| MAE | 56.9 ± 1.3 | 59.5 ± 1.0 | 58.5 ± 1.2 | 58.1 ± 1.1 | 85.6 ± 0.8 | 87.8 ± 0.7 | 87.5 ± 0.9 | 86.7 ± 0.8 |
| DINOv2 | 55.7 ± 1.4 | 57.6 ± 1.1 | 56.8 ± 1.3 | 56.0 ± 1.5 | 79.4 ± 1.0 | 81.5 ± 0.9 | 80.4 ± 1.2 | 79.4 ± 1.1 |
| Tip-Adapter | 56.3 ± 1.2 | 58.4 ± 1.0 | 57.8 ± 1.1 | 56.9 ± 1.3 | 82.2 ± 0.8 | 84.5 ± 0.9 | 83.7 ± 0.9 | 82.9 ± 1.0 |
| AdaptFormer | 59.8 ± 1.1 | 62.1 ± 0.9 | 61.2 ± 1.0 | 60.4 ± 1.2 | 83.9 ± 0.7 | 85.9 ± 0.8 | 85.2 ± 0.9 | 84.3 ± 1.0 |
| LoRA | 58.2 ± 1.3 | 60.2 ± 1.0 | 59.4 ± 1.1 | 58.5 ± 1.4 | 84.4 ± 0.7 | 86.7 ± 0.6 | 85.9 ± 0.8 | 85.1 ± 0.9 |
| **DCRM-ViT** | 60.4 ± 1.0 | 63.4 ± 0.8 | 62.7 ± 0.9 | 58.3 ± 1.3 | 86.2 ± 0.7 | 89.2 ± 0.6 | 88.6 ± 0.8 | 88.1 ± 0.9 |

(b) Segmentation Dice (↑)

| Method | Ultrasound | | | Cardiac CT/MRI | | |
|---|---|---|---|---|---|---|
| | BUS-UCLM | BUID | BUS-BRA | ACDC | MMWHS–CT | MMWHS–MRI |
| U-Net | 0.682 | 0.651 | 0.723 | 0.888 | 0.825 | 0.793 |
| SAMed | 0.732 | 0.702 | 0.668 | 0.902 | 0.837 | 0.805 |
| FetalCLIP | 0.781 | 0.742 | 0.719 | 0.894 | 0.849 | 0.818 |
| SAMUS | 0.817 | 0.774 | 0.802 | 0.905 | 0.861 | 0.831 |
| **DCRM-ViT** | **0.862** | **0.789** | **0.834** | **0.928** | **0.880** | **0.856** |

**Ablation Study.** Our ablation studies evaluated the impact of specific architectural changes in the `DCRM-ViT` model, focusing on fetal datasets (Fpus23 and Fetal Planes). The results for the experiment without and with some components of the `DCRM-ViT` are shown in Table 6. Similarly, increasing the number of adapter layers consistently improved performance on fetal datasets until a size of 12, as can be seen in Table 9. We also perform the ablation for the backbone used (Table 9), as well as the impact of epoch size on the performance, as presented in Table 7a. The performance here tends to decrease after 30 epochs due to the overfitting problem. More ablations are also shown in A.11 and A.12.

# 5 CONCLUSION

This paper introduced `DCRM-ViT`, a transformer-based framework that embeds per-sample domain-aware adaptation to improve image classification and segmentation across medical and general domains. Results show that `DCRM-ViT` surpasses traditional models like CLIP and DINOv2 in adaptability and accuracy. Including `RMB` layers has notably enhanced `DCRM-ViT`'s generalization capabilities, especially in medical imaging, leading to significant performance improvements. Ablation studies affirm the benefits of architectural adjustments such as `RMB` layers and drop path regularization in boosting performance and mitigating overfitting.

## REPRODUCIBILITY STATEMENT.

To facilitate reproducibility, we describe our architecture details along with training setup in Section 3. Besides, we also provided our pseudo-code in A.9 and experimental setup in A.8. We will release code, configs, and pretrained DCRM weights to reproduce all results end-to-end after acceptance.

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

# Appendix for DCRM-VIT

SMALL CAPS: APPENDIX CONTENTS

# A APPENDIX

## A.1 LLM USAGE STATEMENT

We made limited use of large language models to enhance the clarity and readability of the text. They were not involved in the conception of ideas, experiment design, analysis, or the production of results.

Table 8: Comparison of module types in the `DCRM-ViT` framework

| Module Type | Primary Role | Input Sensitivity | Mechanism | Benefit |
|---|---|---|---|---|
| `DR.` | Domain-aware modulation | Varies per input (e.g., fetal vs. general images) | Gate-channel architecture with `PSN`-generated parameters | Enables input-dependent adaptation across domains |
| `RMB` | Task-specific fine-tuning | Fixed per task | Lightweight residual bottleneck modules | Enables parameter-efficient customization and improved classification performance |

## A.2 ZERO-SHOT TRANSFORMATION

This adaptation process is formulated as an optimization problem that aims to minimize the squared euclidean distance between the transformed feature representations of the source models and the corresponding CLIP embeddings, with an added regularization term to mitigate overfitting. Here, the source model corresponds to the model that lacks direct zero-shot capability. The mathematical expression for this optimization function includes a Frobenius norm of $W$ as the regularization term:

$$\min_{W,b} \left\{ \sum_{i=1}^{N} \|W f_{\text{source}}(\mathbf{x}_i) + b - f_{\text{CLIP}}(\mathbf{x}_i)\|_2^2 + \lambda \|W\|_F^2 \right\} \quad (5)$$

Where $\lambda$ is a regularization parameter that helps balance the fit and complexity of the model to enhance generalization. The transformation $W$ and bias $b$ are updated iteratively using stochastic gradient descent, with the update rules defined as:

$$W^{(t+1)} = W^{(t)} - \eta \left( \nabla_W L(W, b; \mathbf{x}^{(t)}) + \lambda W^{(t)} \right) \quad (6)$$

$$b^{(t+1)} = b^{(t)} - \eta \nabla_b L(W, b; \mathbf{x}^{(t)}) \quad (7)$$

$$L(W, b; \mathbf{x}) = \sum_{i=1}^{n} \|W f_{\text{source}}(\mathbf{x}_i) + b - f_{\text{CLIP}}(\mathbf{x}_i)\|_2^2 \quad (8)$$

Here, $\eta$ denotes the learning rate and $t$ represents the iteration number. After training, the adaptation's effectiveness is quantified by computing the cosine similarity between the transformed feature vectors and the CLIP embeddings. This metric provides a measure of how well the adapted model features align with the multimodal semantic space of CLIP:

$$\text{cosine\_similarity} = \frac{\langle W f_{\text{source}}(\mathbf{x}), f_{\text{CLIP}}(\mathbf{x}) \rangle}{\|W f_{\text{source}}(\mathbf{x})\| \|f_{\text{CLIP}}(\mathbf{x})\|} \quad (9)$$

A high cosine similarity indicates successful alignment, affirming the model's capability to perform zero-shot classification by interpreting textual descriptions associated with images. This enables these traditionally non-zero-shot models to recognize and categorize images without explicit prior training on specific class labels. This adaptation extends the utility of these models for advanced applications where labels are scarce or unavailable, enhancing their applicability in diverse real-world scenarios.

## A.3 LIMITATIONS AND FUTURE WORK

The granularity of `DCRM-ViT`'s `DR.` module may not be sufficient for complex medical scenarios, as it primarily distinguishes between broad image categories without delving into finer medical subdomains. This limits its diagnostic capabilities in environments that require detailed differentiation among medical conditions. Moreover, `DCRM-ViT`'s performance heavily depends on the quality of its underlying pre-trained backbone ViT models. By having inadequate or non-representative pre-training, the model's effectiveness can be affected in specialized medical tasks, affecting its

adaptability and precision. Moreover, our evaluation focuses on static 2D scans which shows that **DCRM-ViT** has not yet been tested on video sequences or other medical imaging modalities like X-ray. Finally, **DCRM-ViT** is based on classification and segmentation tasks, whereas many clinical applications (e.g., landmark detection) require further dense predictions.

To refine **DCRM-ViT**, we aim to enhance the ability of the **DR.** module to recognize and adapt to more specific medical modalities like PET, X-Ray, and other scans, which will cover a broad spectrum of adaptability in the medical domain. Additionally, the current structure of **DCRM-ViT** follows a linear approach, where we tend to explore other parallel options to combine the different modules, like having separate **RMB** branches (e.g., one branch for natural features, one for the MRI domain, and one for fetal), followed by a fusion mechanism. Furthermore, we will generalize the **DR.** + **RMB** integration for dense prediction by embedding domain-aware calibration directly into landmark-detection heads, thereby enabling pixel-level refinement of fetal structures. Finally, to enable video analysis, we also look forward to incorporating temporal feature fusion, for example, via a lightweight spatio-temporal transformer or recurrent memory tokens to enforce consistency across video frames and improve robustness to motion blur.

## A.4 Negative Societal Impact

While **DCRM-ViT** presents a promising advancement for enhancing prenatal diagnostics, its implementation may inadvertently introduce certain negative societal consequences. A primary concern is the risk of overreliance on automated model output, which could reduce hands-on diagnostic practice and experiential learning opportunities for clinicians. This reduction in active engagement, especially among novice practitioners, risks disrupting the essential expert-novice developmental trajectory. Such disruption may impair the acquisition of diagnostic intuition and decision-making skills critical for clinical expertise. Beane (Beane, 2024) emphasizes that preserving human skill in an era of intelligent machines requires careful balancing to avoid the premature automation of tasks that undermine the development of expertise and the nuanced judgment that only hands-on experience can cultivate.

Moreover, since the model is trained on data from specific ultrasound devices and patient populations, domain shifts may degrade model performance when applied across different scanners or diverse demographic groups. This limitation highlights the imperative for ongoing domain adaptation, validation, and recalibration using heterogeneous datasets to ensure consistent accuracy and fairness across clinical contexts.

Finally, despite adherence to rigorous privacy protocols, medical imaging systems inherently carry a residual risk of unauthorized data access or breaches. Given the sensitivity of prenatal imaging data, robust adversarial safeguards and comprehensive cybersecurity measures are critical to mitigate such vulnerabilities and uphold patient confidentiality.

## A.5 Key Insights.

There is a notable difference in the zero-shot performance between fetal and natural image datasets. However, it is worth noting that this accuracy has been reported in zero-shot settings while using the feature transformation approach (Sec. 3). This lower performance can be improved further by using several targeted improvements, like expansion of training data to cover more variations of the medical domain, implementation of a contrastive pre-training approach specifically for the fetal domain, and modifying the transformation approach by using CLIP embedding space tailored for the fetal domain rather than the natural imagery domain. Moreover, during the feature transformation, we used the normal CLIP embedding space, which limits the capability of **DCRM-ViT**.

## A.6 Extended Description of Methodology

In this sub-section, we provide further details for the proposed methodology of **DCRM-ViT** model.

### A.6.1 Soft Parameter Sharing

We implement a soft-parameter sharing mechanism to effectively utilize knowledge from both domains. The final **RMB** parameters are computed by weighting the domain-specific parameters

according to the domain probabilities. This can be mathematically written as $\theta_A = p_{\text{medical}}\theta_{A,\text{medical}} + p_{\text{natural}}\theta_{A,\text{natural}}$. This approach enables the model to handle images with features from different domains or ambiguous characteristics, thus enhancing its robustness and generalization.

### A.6.2 REGULARIZATION OF DYNAMIC PARAMETERS

To prevent the dynamically generated parameters and domain attention biases from deviating excessively from the base parameters, we introduce a regularization term in the loss function as $\mathcal{L}_{\text{reg}} = \lambda \left( \|\theta_A - \theta_{A,\text{base}}\|^2 + \|\mathbf{B}_d - \mathbf{B}_{\text{base}}\|^2 \right)$. Here, $\theta_{A,\text{base}}$ and $\mathbf{B}_{\text{base}}$ are the base **RMB** parameters and attention biases (initialized from the pre-trained weights of ViT), and $\lambda$ is a regularization coefficient. This regularization ensures that the model retains the foundational knowledge from the pre-trained model while adapting to domain-specific nuances.

### A.6.3 DROP PATH REGULARIZATION

Drop path regularization is implemented to enhance the generalization capability of the **DCRM-ViT** model and prevent overfitting. This technique randomly drops Transformer blocks during the training phase, encouraging the model to develop redundant paths for information processing. This increases its fault tolerance and reduces dependence on any single path during inference. The drop path regularization can be mathematically expressed as:

$$\mathbf{h}_{\text{drop}} = \begin{cases} \mathbf{h} & \text{with probability } (1-p) \\ 0 & \text{with probability } p \end{cases} \tag{10}$$

where $p$ is the drop path rate.

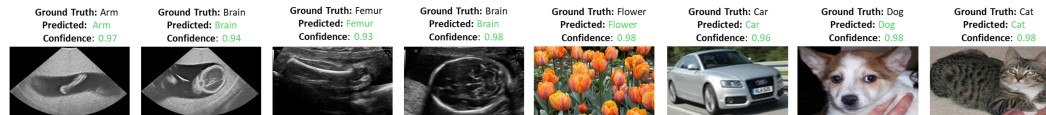

Figure 4: **DCRM-ViT** results using ultrasound and natural images, showing labels and confidence levels

### A.6.4 ROBUSTNESS TO NOISE.

The robustness of domain probabilities to noise in medical data is a crucial concern due to the inherently noisy nature of such data. Here, the domain classifier is trained not only on clean images but also on medical images, learning the discriminative nature between both domains. In case we remove the noise completely, then the model will probably get overfit and will not be able to perform well when shown some real-time clinical medical images, which definitely would contain the noise. So, the model should be adapted to the nature of the medical domain accordingly. Therefore, we have already incorporated the Drop Path (as explained above) and regularization approach within the domain classifier to prevent overfitting to noisy or outlier data points, enhancing the generalization capabilities of the **DR.** module across noisy inputs.

### A.7 QUALITATIVE RESULTS

Figure 4 illustrates the correctly classified examples from a batch of test data spanning both fetal-ultrasound (arm, brain, femur) and natural-image (flower, car, dog, cat) domains. In all cases, the model's top-1 prediction matches the ground truth with high confidence (0.93–0.98), demonstrating robust domain-agnostic recognition that complements the quantitative gains reported earlier.

### A.8 EXPERIMENTAL SETUP

The experiments are conducted on NVIDIA A100 GPU with 40 GB of VRAM and 128 GB of RAM. Adam (Kingma & Ba, 2014) optimizer was used for all model training, whereas SGD (Ruder, 2016), along with the CosineAnnealingLR scheduler, was used to adapt the self-supervised models to

zero-shot configurations. We used different models like MAE (He et al., 2022), DINOv2 (Oquab et al., 2023), and CLIP (Radford et al., 2021), as well as the low-rank methods like Tip-Adapter (Zhang et al., 2021), AdaptFormer (Chen et al., 2022), and LoRA (Hu et al., 2021). The batch size of 32 was used for all the datasets. We use the inner-loop learning rate to be relatively high with value of $1 \times 10^{-3}$ so that each **RMB** can rapidly adapt to its task-specific data. In contrast, we use a bit lower learning rate of $1 \times 10^{-4}$ as the outer loop updates the parameters more conservatively.

For validation and performance testing, a split of 60-20-20 is used, where 60% of the data from each dataset is used for training ($\mathcal{D}_T^{\text{dom}}$, $\mathcal{D}_T^{\text{task}}$), 20% for validation, and the remaining 20% for testing. The difference between $\mathcal{D}_T^{\text{dom}}$ and $\mathcal{D}_T^{\text{task}}$ is that the former is used as a binary classification problem where the classes are either fetal or natural domain. In contrast, the latter refers to the main task classes like fetal arm, head, abdomen, etc. We evaluate the performance of each model using two primary metrics: accuracy and F1 score. Accuracy measures the proportion of correct predictions made by the model out of all predictions. In contrast, the F1 Score assesses the model's precision and recall balance, which is particularly useful in scenarios with imbalanced datasets.

### A.9 PSEUDO ALGORITHM

---

**Algorithm 1** Training Procedure for Model Adaptation

---

1: **Initialization:** Start with pre-trained transformer parameters $\theta$, initial **RMB** parameters $\phi_{\text{base}}$, **DR.** module parameters $\omega_{\text{base}}$, and learning rates $\alpha$ and $\beta$.
2: **for** each task $T$ **do**
3:     **Domain Probability Estimation:** Use the domain classifier $D$ to compute $D(\mathbf{x})$ for images in $\mathcal{D}_T^{\text{task}}$ and $\mathcal{D}_T^{\text{dom}}$.
4:     **Inner Loop:**
5:         Generate **RMB** parameters $\phi$ using the **PSN** unit and domain probabilities.
6:         Fine-tune $\phi$ on $\mathcal{D}_T^{\text{task}}$.
7:         Update $\phi'_T$ for task $T$.
8:     **Outer Loop:**
9:         Compute $\mathcal{L}_{\text{total}}$ on $\mathcal{D}_T^{\text{dom}}$.
10:        Update $\omega$, $\phi$, and $\alpha$.
11: **end for**
12: **Repeat:**
13:     Iterate over all tasks, performing inner and outer loop updates until convergence.

---

### A.10 DATASETS

This paper incorporates ultrasound, CT, and MRI, as well as natural imagery datasets to comprehensively evaluate the effectiveness of the models. Further details for each modality is provided below.

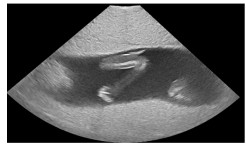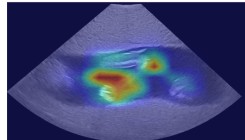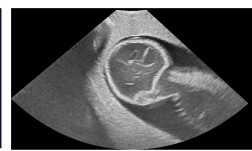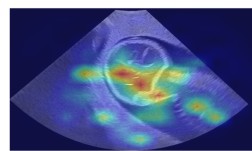

Figure 5: Visualization of attention maps

#### A.10.1 ULTRASOUND DATASETS

**FPUS23.** The FPUS23 dataset (Prabakaran et al., 2023) utilizes a simulated 23-week gestation fetus phantom to overcome the ethical challenges associated with actual patient data. This dataset, comprising 15,728 ultrasound images captured with the Philips Epiq-7 system, is enhanced by Anatomically Intelligent Ultrasound (AIUS) technology. It includes images categorized into four classes: Head, Abdomen, Arms, and Legs, demonstrating its diversity and applicability for anatomical

studies. We filtered the images from this dataset so that each image would have a single class like arm or an abdomen, and then applied various augmentations to increase the size of the dataset.

**Fetal Planes DB.** The Fetal Planes (FP) dataset (Burgos-Artizzu et al., 2020) features over 12,400 annotated ultrasound images from 1,792 patients. These images are sorted into six categories reflecting key fetal anatomical planes used in prenatal screenings: Abdomen, Brain, Femur, Thorax, and the maternal cervix. Additional categorization into sub-planes, including trans-thalamic, trans-cerebellum, and trans-ventricular, provides detailed insights into brain anatomy, although these are not the primary focus of this paper.

**BUS-UCLM.** (Vallez et al., 2025) is a curated set of 683 breast ultrasound images from 38 patients acquired at Ciudad Real General University Hospital on a Siemens ACUSON S2000 with an 18L6 HD probe. Each image has an expert RGB mask (black = normal, green = benign, red = malignant) and having a total class count of 419 normal, 174 benign, 90 malignant samples, respectively.

**BUID.** (Al-Dhbyani et al., 2020) is a collection of 780 PNG images ($500 \times 500$ px) from $\tilde{6}00$ female patients collected at Baheya Hospital (Cairo) using LOGIQ E9 / LOGIQ E9 Agile systems. It contains normal, benign, and malignant classes with per-image freehand mask ground truths. It has a class count of 487 benign, 210 malignant, and 133 normal, respectively.

**BUS-BRA.**(Gómez-Flores et al., 2024) is a public dataset of 1,875 anonymized breast ultrasound images from 1,064 female patients acquired on four scanners at Brazil's National Institute of Cancer. It provides biopsy-proven labels (722 benign, 342 malignant), expert lesion masks, and standardized 5- and 10-fold cross-validation splits.

### A.10.2  CT/MRI DATASETS

**ACDC.** (Bernard et al., 2018) is a cine-MRI dataset of 150 patients evenly split into five diagnostic groups (NOR, MINF, DCM, HCM, ARV), acquired over six years on 1.5T/3T Siemens scanners with SSFP sequences. The training set includes 100 labeled subjects, and the test set contains 50. Expert annotations are provided at the ED / ES for the LV / RV cavities and the LV myocardium.

**MMWHS.** (Zhuang et al., 2019) is a challenge dataset that provides 120 clinical 3D cardiac volumes (60 CT and 60 MRI) covering the whole heart. It targets segmentation of standard heart substructures and enables consistent benchmarking across modalities.

### A.10.3  NATURAL IMAGERY DATASETS

For natural-image evaluation, we use CIFAR-10 Krizhevsky et al. (2009), Caltech-101 Fei-Fei et al. (2004), Natural Images Roy et al. (2018), Food-101 Bossard et al. (2014), SUN397 Xiao et al. (2010), and Stanford Cars Krause et al. (2013).

### A.11  ABLATION FOR **RMB** DEPTH

Table 9: Compact ablation on **RMB** depth (left block) and backbone choice (right block) in **DCRM-ViT**. The left value shows accuracy while right value corresponds to F1 score.

| Dataset | # RMB layers | | | Backbone | | |
|---|---|---|---|---|---|---|
| | 8 | 12 | 14 | ResNet-50 | VGG-16 | ViT-B/16 |
| FPUS23 | 61.4±1.3 / 0.61±0.04 | 63.4±0.9 / 0.69±0.03 | 62.7±1.1 / 0.65±0.04 | 60.5±1.5 / 0.61±0.05 | 59.2±1.2 / 0.60±0.03 | 63.4±0.8 / 0.62±0.02 |
| Fetal Planes | 87.2±0.7 / 0.87±0.02 | 89.2±0.6 / 0.90±0.01 | 88.2±0.8 / 0.88±0.03 | 87.0±0.9 / 0.87±0.04 | 85.5±1.0 / 0.86±0.05 | 89.2±0.5 / 0.88±0.02 |

### A.12  ABLATION FOR DROP PATH RATE

To enlighten the impact of drop path regularization on the performance of the **DCRM-ViT** model, we conducted three experiments across two datasets, FPUS23 and FP. The results, as delineated in Table 10, illustrate that a moderate drop path rate of 0.1 enhances both accuracy and F1 scores, particularly on the FP dataset. Conversely, increasing the drop path rate to 0.3 decreased performance metrics, underscoring the critical need for precise tuning of regularization parameters to maintain the delicate balance between model robustness and learning efficiency.

Table 10: Impact of drop path regularization on model performance across FPUS23 and FP datasets.

| Drop Path Rate | FPUS23 | | Fetal Planes | |
|---|---|---|---|---|
| | Accuracy (%) | F1 Score | Accuracy (%) | F1 Score |
| 0.0 | 63.1 ± 1.4 | 0.67 ± 0.05 | 88.5 ± 0.7 | 0.88 ± 0.02 |
| 0.1 | 63.4 ± 0.8 | 0.69 ± 0.03 | 89.0 ± 0.4 | 0.89 ± 0.01 |
| 0.2 | 62.9 ± 1.2 | 0.68 ± 0.06 | 89.3 ± 0.9 | 0.90 ± 0.03 |
| 0.3 | 62.3 ± 1.0 | 0.67 ± 0.04 | 88.7 ± 0.5 | 0.89 ± 0.04 |

