# OpenReview forum: "DCRM-ViT: Domain Conditioned Residual Modulation for Multi-Domain Vision Transformers"
_ICLR.cc/2026/Conference — ICLR 2026 Conference Withdrawn Submission_

### Official Review · Reviewer_gei2 · 2025-10-21

**Soundness:** 2
**Presentation:** 2
**Contribution:** 3
**Rating:** 4
**Confidence:** 4

**Summary:**

This paper proposes DCRM-ViT, a domain-conditioned residual modulation framework designed to adapt vision transformers efficiently across different domains. It includes lightweight residual modulation blocks (RMB), whose parameters are generated by a domain router (DR.) and parameter synthesizer network (PSN). DCRM-ViT is optimized through a bi-level scheme: a short inner loop to update RMB for task-specific learning and an outer loop to update DR., PSN, and RMB's initializations/step size for domain-specific learning.

The proposed model is evaluated on classification and segmentation datasets, and compared against prior foundation models and parameter-efficient fine-tuning (PEFT) methods.

**Strengths:**

* S1: This paper explores multi-domain vision transformers and PEFT, which is of interest to the community.

* S2: The integration of domain router with learnable task-specific modules, RMB, isreasonable.

* S3: The proposed approach shows improved classification and segmentation performance compared with baseline methods presented in the paper.

**Weaknesses:**

**Major Weakness**

* W1: The experimental evaluation is limited. DCRM-ViT is designed for multi-domain adaptation, but it is not compared with multi-domain or multi-task methods, such as domain-specific normalization, TAPS [R1], and AdaMV-MoE[R2].

* W2: Comparisons are restricted to 3 a little older PEFT methods, which weakens the empirical support for the claimed advantages. Comparing with newer baselines would make the paper stronger.

* W3: The experimental settings of baselines are unclear. Are they trained jointly on multiple domains/tasks jointly like DCRM-ViT or separately per domain? Without this information, fairness and validity of comparisons are uncertain.  In addition, presenting linear-probing results of foundation models would help clarify whether improvements come from better representation learning or overfitting to small datasets.

* W4: Paper writing and methodological presentation need to be improved.  Currently it is hard to distinguish the authors' methodology contributions from module-by-module explanations in the method section.

[R1] Wallingford, Matthew, et al. "Task adaptive parameter sharing for multi-task learning." CVPR 2022.

[R2] Chen, Tianlong, et al. "Adamv-moe: Adaptive multi-task vision mixture-of-experts." ICCV 2023.

**Questions:**

**Primary  Questions/Suggestions**

* QS1: Could the authors include comparisons with SOTA multi-domain and PEFT approaches, as well as linear-probing results, to better validate the effectiveness of DCRM-ViT?

* QS2: Could the authors clarify the training setups for baselines?

* QS3: Consider refining the writing for clarity, particularly in the methodology section

---

### Official Review · Reviewer_eQvq · 2025-10-31

**Soundness:** 3
**Presentation:** 3
**Contribution:** 2
**Rating:** 4
**Confidence:** 3

**Summary:**

This work provides a novel architecture for medical AI domains. The problem the authors focus on is poor performance of models finetuned on medical data when tested on natural image domains. To address the issues of catastrophic forgetting and non-interpretable weight changes, they suggest an architecture that learns domain weights from input features and uses synthesizers to map these weights to residuals and attention biases that allow the architecture to meta-learn strategies for generalizing across domains.

**Strengths:**

Complex architecture, so great to have Figure 3 highlighting all the ingredients.

Evaluation against prior work is thorough and performance boost gained by this method is clearly shown, with error bars.

Thorough ablations of each piece of the proposed architecture.

**Weaknesses:**

Motivation: Would like to see more explanation of why the cross-domain transfer you are targeting is from a model finetuned on a medical modality to natural images? In what practical applications do we need our medical models to still perform well on natural images? Are we assuming we have a generalist medical AI model that we want to be adaptive to other domains? Is the focus on natural images a proxy for generalization to perhaps other medical imagining modalities? If so, why are natural image dataset results a fair substitution? Why not do cross-domain generalization tests on other medical modalities?

For table 4, why are cross-domain results not compared against all related works, as previous tables do?

**Questions:**

See previous sections. Open to changing my score based on responses to these questions!

---

### Official Review · Reviewer_7rsA · 2025-11-01

**Soundness:** 2
**Presentation:** 2
**Contribution:** 2
**Rating:** 4
**Confidence:** 2

**Summary:**

This paper introduces DCRM-ViT, a Vision Transformer framework designed to handle both medical imaging (ultrasound, CT, MRI) and natural images within a single model. The key contribution is a domain-conditioned residual modulation approach that keeps the backbone ViT frozen while adding lightweight Residual Modulation Blocks (RMB) whose parameters are dynamically generated per-sample by a Domain Router (DR) and Parameter Synthesizer Network (PSN). The training employs bi-level optimization: an inner loop adapts RMB parameters for task-specific learning, while an outer loop meta-learns the domain routing and parameter synthesis components. The method shows improvements of 1-3 percentage points over baselines across medical and natural image datasets while using only 3.3M trainable parameters.

**Strengths:**

1. **Comprehensive evaluation**: Thorough experiments across multiple medical modalities (ultrasound, CT, MRI) and natural image datasets
2. **Parameter efficiency**: Achieves reasonable performance with only 3.3M trainable parameters vs. full fine-tuning
3. **Practical relevance**: Addresses a real clinical need for models that work across medical and natural domains
4. **Cross-domain transfer**: Demonstrates maintained performance when switching between domains
5. **Ablation studies**: Provides systematic ablations of architectural components

**Weaknesses:**

1. **Limited technical novelty**: The paper combines existing techniques without significant innovation. Domain adaptation through adapters, parameter generation networks, and bi-level optimization have all been explored previously.

2. **Oversimplified domain modeling**: The binary medical/natural domain distinction is too coarse. Medical imaging encompasses diverse modalities with vastly different characteristics that the current approach doesn't address.

3. **Questionable zero-shot evaluation**: The transformation approach (Eqs. 5-9) to enable "zero-shot" capabilities for non-CLIP models involves training on CLIP embeddings, which isn't true zero-shot learning.

4. **Insufficient analysis**: The paper lacks visualization of learned domain weights, analysis of failure cases, or interpretation of what the domain router actually learns.

5. **Missing computational analysis**: While parameter counts are reported, actual training/inference time overhead of the bi-level optimization and dynamic parameter generation is unclear.

6. **Statistical rigor**: No significance testing despite small margins; confidence intervals in tables appear to be standard deviations rather than proper confidence intervals.

**Questions:**

1. How does the domain router handle ambiguous images with mixed domain characteristics? Can you provide examples and domain weight visualizations?

2. What is the actual wall-clock inference time overhead compared to the frozen backbone? The dynamic parameter generation seems computationally expensive.

3. Can this approach scale beyond binary domain classification? How would it handle multiple fine-grained medical subdomains?

4. The bi-level optimization seems complex - how sensitive is it to hyperparameter choices? What are the convergence properties?

5. Could you clarify the zero-shot transformation methodology? Training on CLIP embeddings seems to invalidate the zero-shot comparison.

---

### Official Review · Reviewer_ySsg · 2025-11-07

**Soundness:** 3
**Presentation:** 2
**Contribution:** 2
**Rating:** 4
**Confidence:** 3

**Summary:**

This paper proposes DCRM-ViT, a model that preserves general capability of pre-trained vision foundation models while adapting to new medical domain data. Specifically, DCRM-ViT keeps the backbone frozen and augments each block with synthesized parameters. A meta-learning framework is adopted for optimization. Experiments on both natural image and medical image data show superior performances compared to vanilla vision foundation models and lightweight adaptation method.

**Strengths:**

The proposed DCRM-ViT method achieves strong performances across diverse datasets with only a small number of trainable parameters and small compute cost. The idea of keeping base model frozen while introducing a small number of extra parameters has been well studied in the LoRA series of work, and is expected to work when adapting to new domains, but generating these parameters on-the-fly is quite novel and effective.

**Weaknesses:**

1. Overall, I get that the authors introduced different modules like Residual Modulation Block (RMB), Domain Router (DR) and Parameter Synthesizer Network (PSN) to adapt the model to new medical domain data, but I fail to grasp the high-level idea and intuition behind those designs by reading the paper. Right now they just look a bunch of new modules inserted into existing models and magically work.

2. The evaluation on natural image datasets is limited. The authors are encouraged to check https://github.com/LAION-AI/CLIP_benchmark for a complete set of benchmarks. Evaluation on only CIFAR-10 and Caltech101 only covers a small number of natural image distribution.

3. The drawing of figure 3 is particularly messy and confusing. I highly suggest that the authors rearrange the legend part, as now it is hard to draw correspondence between the legends and the flow chart.

**Questions:**

1. How does DCRM-ViT benefit from the MAML-style meta learning? The authors claim that "Without this nested scheme, the PSN unit would receive conflicting gradient signals" (line 310), yet I cannot find any related experimental proof. Also, why does training on all domains and tasks lead to inferior performance? The authors are encouraged to give an intuitive explanation.

2. Similarly, I do not see any ablation study about the design choice. For instance, why is generating parameters on-the-fly better? Why use a soft-parameter sharing mechanism. And there are many more. The authors are encouraged to give both intuitive explanation as well as experimental results on this matter.

3. What is the base foundation model for DCRM-ViT? I cannot find any related details, though I suppose it is likely to be a CLIP model variant?

---

### Note · Authors · 2025-11-14

I have read and agree with the venue's withdrawal policy on behalf of myself and my co-authors.